# Challenges in Quantifying 8-OHdG and 8-Isoprostane in Exhaled Breath Condensate

**DOI:** 10.3390/antiox11050830

**Published:** 2022-04-25

**Authors:** Maud Hemmendinger, Jean-Jacques Sauvain, Nancy B. Hopf, Guillaume Suárez, Irina Guseva Canu

**Affiliations:** Center for Primary Care and Public Health (Unisanté), Department of Occupational and Environmental Health, 1066 Lausanne, Switzerland; jean-jacques.sauvain@unisante.ch (J.-J.S.); nancy.hopf@unisante.ch (N.B.H.); guillaume.suarez@unisante.ch (G.S.); irina.guseva-canu@unisante.ch (I.G.C.)

**Keywords:** 8-hydroxy-2′-deoxyguanosine, 8-Isoprostane, oxidative stress biomarkers, exhaled breath condensate, liquid chromatography, mass spectrometry

## Abstract

Exhaled breath condensate (EBC) has attracted substantial interest in the last few years, enabling the assessment of airway inflammation with a non-invasive method. Concentrations of 8-Hydroxydesoxyguanosine (8-OHdG) and 8-isoprostane in EBC have been suggested as candidate biomarkers for lung diseases associated with inflammation and oxidative stress. EBC is a diluted biological matrix and consequently, requires highly sensitive chemical analytic methods (picomolar range) for biomarker quantification. We developed a new liquid chromatography coupled to tandem mass spectrometry method to quantify 8-OHdG and 8-isoprostane in EBC simultaneously. We applied this novel biomarker method in EBC obtained from 10 healthy subjects, 7 asthmatic subjects, and 9 subjects with chronic obstructive pulmonary disease. Both biomarkers were below the limit of detection (LOD) despite the good sensitivity of the chemical analytical method (LOD = 0.5 pg/mL for 8-OHdG; 1 pg/mL for 8-isoprostane). This lack of detection might result from factors affecting EBC collections. These findings are in line with methodological concerns already raised regarding the reliability of EBC collection for quantification of 8-OHdG and 8-isoprostane. Precaution is therefore needed when comparing literature results without considering methodological issues relative to EBC collection and analysis. Loss of analyte during EBC collection procedures still needs to be resolved before using these oxidative stress biomarkers in EBC.

## 1. Introduction

Chronic exposure to airborne particles, and in particular fine and ultrafine particles, have been shown to induce oxidative stress [1,2]. Oxidative stress is an imbalance in the production of reactive oxygen species (ROS) and has been suggested as a potential mechanism for inflammation, apoptosis, and genotoxicity, among others, leading to adverse health outcomes [3]. ROS induce cellular release of inflammatory mediators that can result in tissue damage, epigenetic changes, protein alteration, lipid peroxidation, structural DNA damage, remodeling of extracellular matrix, and ultimately, lead to respiratory diseases [4]. Exhaled breath condensate (EBC) has been suggested as a simple and non-invasive sampling procedure, and has been used to diagnose pulmonary pathologies as it directly monitors the airway inflammation and oxidative stress in the lungs (target organ) [5]. 

EBC contains aerosolized airway epithelial lining fluid and particles trapped from inhaled air. These particles, due to their chemical composition, can locally exert pro-inflammatory effects with the generation of exogenous oxidative stress substances. The EBC also contains endogenous volatile and non-volatile substances released into the lining fluid. Some of these substances have the potential to be used as biomarkers for oxidative stress. Measuring biomarkers in EBC has been proposed as a biomonitoring tool of occupationally exposed workers [6]. Validation is a prerequisite condition for the successful development and use of biomarkers [7]. 

The intrinsic characteristics of a biomarker (accuracy, sensitivity, and specificity), its analytical precision, and its pathophysiological significance are the most important properties that best validate a biomarker for its use in clinical practice [8]. Oxidative biomarkers are often detected at picomolar concentrations in biological media such as EBC, which warrant very sensitive and selective analytical methods as well as careful sample preparations. In this respect, significant effects of interferences [9] and lack of reproducibility or loss of analyte during EBC collection procedures [10,11,12] are reported in the scientific literature. These major methodological heterogeneities could inhibit the potential clinical use for EBC unless robust analytical methods are used considering all confounding factors and sources of measurement error and bias [5,9].

One of the major physiological mediators to quantify oxidative stress damage in vivo are F2-isoprostanes. They correspond to a class of prostaglandins, resulting mainly from non-enzymatically peroxidation of arachidonic acid in membrane phospholipids [13]. A minor pathway for prostaglandin formation is via a cyclooxygenase pathway from human platelets and monocytes [14]. The most prevalent F2-isoprostane in humans is 8-iso-15(S)-prostaglandin F2α, also known as 8-isoprostane. Due to its stability and specificity for lipid peroxidation, 8-isoprostane is postulated to be a reliable biomarker for lipid peroxidation and represent a quantitative measure of oxidative stress [5,15]. Compared to controls, increased 8-isoprostane concentrations were reported in EBC of patients with both stable and exacerbated chronic obstructive pulmonary disease (COPD) [16], cystic fibrosis [17], and asthma [10,18,19]. 

Substantial DNA damage in lungs can occur during oxidative stress, with the easiest oxidized base being guanosine [20]. 8-Hydroxy-2′-deoxyguanosine (8-OHdG) is considered the most frequently detected and studied oxidized DNA product [21]. DNA damages are usually repaired via the base excision repair pathway and oxidized DNA products are spontaneously released from cells or due to necrosis and apoptosis, leading to increased levels of circulating cell-free oxidized DNA. The measurement of 8-OHdG in EBC could thus be a non-invasive approach to understand the molecular pathological changes that occur in cells from lungs and respiratory airways.

Two main analytical approaches are available for 8-OHdG and 8-isoprostane analysis in EBC: immunoassay and chemical analytical methods. Immunoassay-based methods are widely used in experimental and clinical research for 8-OHdG and 8-isoprostane measurements due to their simplicity and low cost. However, large discrepancies between labor intensive chemical analytical methods and immunoassays have been observed [22,23]. These discrepancies are related to the cross-reactivity between isoprostane and structurally similar isomers or biological impurities interfering with antibody binding [24,25], thus rendering the immunoassays less specific. Furthermore, the sensitivity and specificity of immunoassay kits vary considerably from one manufacturer to another [26]. Immunoassays are not yet designed and validated for EBC, but could potentially be used if possible matrix effects are formally studied. Most importantly, immunoassays are indirect methods that require validation by reference chemical analytical methods that can unequivocally identify the molecule of interest [27].

Chemical analytical methods such as gas or liquid chromatography with MS detection (GC–MS or LC–MS, respectively) could be used to achieve satisfactory specificity and accuracy. Several authors have successfully validated GC–MS methods for the determination of 8-isoprostane in human EBC [10,11]. However, the method requires an extensive manual sample preparation with a complex two-step derivatization process [11] in order to stabilize the thermally labile chemical functions of 8-isoprostane [28]. Automation of the current analytical process is thus greatly limited [29]. Conversely, LC–MS could be an alternative method as it is performed directly in liquid conditions. Additionally, substantially larger volumes of liquid can be injected into the instrument, improving the sensibility of the analysis.

It is well known that EBC contains large amounts of endogenous matrix components that can lead to a decreased sensitivity and specificity by interference with the MS detection system [30]. Many teams have developed strategies for the selective isolation of 8-isoprostane, using affinity separation [30,31,32]. Nevertheless, extensive sample preparation and high cost associated with such procedure render such approach unsuitable for routine use. 

In this study, we propose an alternative protocol for simultaneous quantification of 8-OHdG and 8-isoprostane in EBC using the LC–MS method without an extensive sample preparation. For biomonitoring purposes, high-throughput and cost-effective analysis are needed [33,34]. Such a method could be interesting for screening large worker populations or for monitoring exposures to ROS-producing agents such as particles.

## 2. Materials and Methods

### 2.1. Reagents–Chemicals

8-OHdG (≥98%) (2-amino-9-[(2R,4S,5R)-4-hydroxy-5-(hydroxymethyl)oxolan-2-yl]-1,7-dihydropurine-6,8-dione) in a solid form was obtained from Merck (Buchs, St. Gallen, Switzerland). [^15^N5]-8-OHdG as internal standard (IS) was obtained from Cambridge Isotope Laboratories (Tewksbury, MA, USA). 8-Isoprostane ((5Z,8β,9α,11α,13E,15S)-9,11,15-trihydroxyprosta-5,13-dien-1-oic acid) in a solution ≥ 95%, and its deuterated IS 8-Isoprostane-d_4_ ((5Z,8β,9α,11α,13E,15S)-9,11,15-trihydroxyprosta-5,13-dien-1-oic-3,3,4,4-d4 acid), were obtained from Cayman Chemical (Ann Arbor, MI, USA). HPLC grade methanol (≥99.9%) was obtained from Merck (Buchs, Switzerland). LC-MS grade solvents; methanol (≥99.95%) and acetonitrile (≥99.9%) were obtained from Carlo Erba Reagents (Chaussée du Vexin, Val de Reuil, France). Butylated hydroxytoluene (BHT) ≥ 99% was obtained from Sigma-Aldrich Produktions GmbH (Steinheim Germany). LC–MS grade acetic acid was obtained from Honeywell (Seelze, Germany). Ultrapure water was produced in our laboratory with a Milli-Q Advantage water purification system (18.2 MΩ.cm at 25 °C, <3 ppb total organic carbon). 

### 2.2. Preparation of Standards 

Stock solutions at 1 mg/mL of 8-OHdG and 1 mg/mL 8-isoprostane in H_2_O/MeOH (8:2) were prepared and stored at −20 °C. A mother stock solution for each analyte (5000 ng/mL) was prepared by diluting 50 µL of the stock solution of 8-OHdG or 8-isoprostane 1 mg/mL with ultrapure water in a volumetric flask of 10 mL. This stock solution was stored at 4 °C. [^15^N5]-8-OHdG at 500 ng/mL was prepared by diluting 10 µL of [^15^N5]-8-OHdG 25 µg/mL in a conical glass vial with screw cap containing 490 µL of ultrapure water. 8-isoprostane d_4_ at 500 ng/mL was prepared by diluting 10 µL of 8-isoprostane d_4_ 100 µg/mL in a glass tube containing 1990 µL of ultrapure water. Final solutions were vortexed and stored at 4 °C. These stock standard solutions can be used at least 3 months without noticeable concentration variation. These different stock solutions were mixed and diluted in ultrapure water to prepare two daily working solutions of 8-OHdG and 8-isoprostane (500 pg/mL each) and an internal standard mix (Mix IS) of [^15^N5]-8-OHdG and 8-isoprostane d_4_ (2.5 ng/mL each). Five calibration standards (final concentrations of 0, 15, 50, 100, and 300 pg/mL) were obtained by diluting the working solutions as described in the Appendix A. 

### 2.3. EBC Samples Preparation

EBC samples used for validation and quality control (QC) were collected from two healthy non-smoking voluntary adult from our laboratory. EBC was collected according to recommendations by the American Thoracic Society/European Respiratory Society Task Force [5]. The EBC collection device (TurboDeccs, Medivac, Parma, Italy) operated at −10 °C and was equipped with a saliva trap [35] and a disposable polypropylene plastic collection system (DECCS 14 ST kit). The volunteers rinsed their mouth with water just before the collection started. Each volunteer sat comfortably, wore a nose-clip, and breathed in through the mouthpiece for 20 min. The collected EBC liquid was transferred into a cryovial (Sarstedt, Nümbrecht, Germany) and stored at −20 °C (for one day for logistical reasons) then transferred to −80 °C until analysis (within 8 months). 

### 2.4. Analytical Procedure

#### 2.4.1. Sample Preparation

A concentration step is necessary due to the low concentrations of these oxidative stress biomarkers in EBC (Section 4). One ml of the standard or EBC sample was introduced in a plastic tube (1.5 mL), followed by 10 µL of BHT 10 mg/mL and 15 µL of Mix IS 2.5 ng/mL. BHT is considered as essential to prevent further ex vivo oxidative formation of 8-isoprostane [36] without impacting the 8-OHdG concentration in EBC [37]. The mixture was then vortexed and put in a concentrator (Speedvac SVC-100H, Savant Instruments Inc., Farmingdale, NY, USA) under vacuum with heat (set at 45 °C) and pressure (range 10–22 mbar). The complete evaporation of the water took 6.5 h. Another concentration step using lyophilization (freeze-drying) (Freezone 1, Labconco, Fort Scott, KS, USA) at 0.06 mbar and <−50 °C at the condenser for 8 h was also tested. With both techniques, the residue was dissolved in 75 µL of water acidified with 0.1% acetic acid (LC-MS grade) (corresponding to a concentration factor of 13.3), vortexed and sonicated for 5 min. The final solution was transferred into a polypropylene vial (300 µL) (Macherey-Nagel AG, Oensingen, Switzerland) before chemical analysis. Standard solutions (concentration of 0, 5, 15, 25, and 50 pg/mL; Appendix A) were included with each series of samples and followed the same procedure. These standards were used for quantification, whereas the non-concentrated solutions (std 0–300 pg/mL, Section 2.2) were used as a reference. Each sample (50 µL) was injected into the LC–MS system. EBC spiked with both analytes at 5 and 50 pg/mL were prepared and served as quality controls (QC). 

#### 2.4.2. Assessment of Type of Sample Concentration Process

Two concentrating methods based either on lyophilization or using vacuum concentration were performed. Experiments consisted of concentrate with five standards solution in water (0–50 pg/mL of each analyte) or five spiked EBC samples (0–50 pg/mL of each analyte) with each technique. Vials of either glass or low-binding plastic used during this concentration step were tested. By plotting the signal obtained for each solution as a function of the concentration, a linear regression characterized by its slope was drawn through these points. We compared these different slopes with the corresponding standard solutions that have not undergone the concentration step.

#### 2.4.3. EBC Material Surface Adsorption and Protein Interferences

Adsorption phenomena on material surfaces have been reported for biomarkers in EBC, particularly for eicosanoids [10,38,39]. We thus used a procedure described by Tufsesson et al. [12] to coat all plastic surfaces with 0.01% Tween 20 (Sigma Aldrich, Burlington, VT, USA) for 30 min. This included all the pieces of the EBC kit (TurboDeccs sample container and tubing). After this treatment, the material was carefully rinsed three times with ultrapure water and then left to dry in an oven at ambient temperature. An EBC collection was then performed on this material by following the same procedure as previously described. An EBC collection was also simulated by placing 2 mL of a 5 pg/mL standard in the collection tube and leaving it in contact with the plastic for 30 min with stirring. The recovery rate compared to what was originally injected was thus assessed. The effect of using glass or plastic containers (low-binding plastic tubes) during the concentration step using lyophilization or Speedvac was also examined.

We tested a centrifugal concentrator system using a membrane of cellulose triacetate with a cut-off of 5000 Dalton (Vivaspin TCA 5kD, Sartorius, VWR, Dietikon, Switzerland) to avoid possible interferences from proteins during the nebulization in the LC–MS system [30,40]. EBC samples was spiked with the two biomarker standards at 15 pg/mL, and centrifuged (Vivaspin at 4000 rpm for 90 min). The filtrate was concentrated to dryness using the vacuum concentrator and the residue dissolved in 0.1% acetic acid as described previously.

#### 2.4.4. Analytical Conditions

The target biomarkers were analyzed with an ultra-high pressure liquid chromatography (LC) system (Dionex Ultimate 3000) equipped with a C18 column (Zorbax Eclipse Plus 2.1 × 100 mm, 1.8 µm, Agilent, Morges, Switzerland) and operated at 50 °C. The injection volume was 50 µL. The solvent gradient (flow rate of 0.25 mL/min) was a combination of eluent A (H_2_O with 0.1% acetic acid) and eluent B (MeOH/ACN 7:3 with 0.1% acetic acid). The LC was operated with the following program: 100% A for 2 min, decreasing to reach 10% over 8 min, held for 4.5 min, then increased to 100% A in 1 min, and held for 6.5 min. A Triple-Stage Quadrupole MS (TSQ Quantiva Thermo Scientific, Reinach, Switzerland) with electrospray ionization (ESI) was used for detection (instrument parameters shown in Table 1). All data acquisition and processing were accomplished using the Thermo Scientific Chromeleon software.

### 2.5. Method Validation and Applicability

#### 2.5.1. Method Validation 

The optimized method was validated by considering linearity, limit of detection (LOD), limit of quantification (LOQ), intra-day and inter-day precisions, recovery, and matrix effects according to FDA/ICH guidelines [41]. The LOD and LOQ were determined by dividing the error on the origin for the calibration standard by the calibration slope. Different pooled EBC samples were used to determine recovery and repeatability. The calibration curve was determined by plotting the peak area ratio standard/IS as a function of the concentration of the added standard. The final concentration of each oxidative stress marker in EBC was calculated based on the calibration curve obtained with the standard treated identically as for the sample.
(1)$${\left[\mathrm{Analyte}\right]}_{\mathrm{EBC}}=\frac{Ssple-B}{A}\times \frac{1}{13.33}$$
where: [Analyte]_EBC_: Concentration of the considered marker (8-OHdG or 8-isoprostane) in EBC [pmol/mL]. *Ssple*: Peak area of the sample/peak area of the IS. *A*: Slope of the calibration curve (Peak area standard/peak area IS) = f[concentration standard]. *B*: Ordinate at the origin of the calibration curve. 1/13.33: Concentration factor, corresponding to the ratio between the initial 1 mL EBC and the final 75 µL of the final sample after vacuum concentrator treatment. 

#### 2.5.2. Study Population

The suitability of the validated method was assessed using EBC samples from 26 workers from the same workplace. These samples were selected from a study sample of 303 Autonomous Parisian Transportation Administration (RATP) workers. The RATP cohort was stratified based on their spirometry results, distinguishing healthy (*n* = 10), asthmatics (*n* = 7), and COPD presenting subjects (*n* = 9) [34]. Informed consent was obtained from all subjects and the study was approved by the French Personal Protection Committees South-Est II (N°2019-A01652 55) and South-Est IV (N°2020-A03103-36). The demographic characteristics of participants are presented in Table 2.

Investigators were kept blind with respect to the health status, which was only revealed at the statistical analysis stage. The hypothesis tested was that workers with asthma or COPD would present higher concentrations of oxidative stress biomarkers in EBC compared to the healthy workers. 

A pulmonologist classified these diseases following the Global Initiative for Obstructive Lung Disease (GOLD) guidelines [42]. A trained occupational physician conducted pulmonary function tests and recorded forced expiratory volume in one second (FEV1) and forced vital capacity (FVC) using a fully-integrated PC-driven spirometer (Easy on-PC System, NDD medical technologies^®^, Andover, MA, USA) according to the American Thoracic Society (ATS)/European Respiratory Society (ERS) recommendations [43]. 

FEV1/FVC ratio from three values on a maximum of eight attempts were then calculated and presented as a ratio of the predicted normal value of 0.7. A reversibility test was performed 15 min after administering four puffs of 100µg of salbutamol (Ventolin inhaler 100µg/dose, GLAXO SMITHKLINE, Brentford, UK) for FEV1/FVC ratio below normal range (<0.7). FEV1/FVC ratio of less than 0.7 in presence of the clinical symptoms such as dyspnea, chronic cough or sputum production were diagnosed as COPD. Asthmatics were diagnosed with a positive bronchodilator reversibility in presence of intermittent variable clinical airflow obstruction and/or an allergic background. 

Food and drinks consumed within three hours before EBC collection were recorded in a standardized form. None of the participants reported drinking coffee within the hour before EBC collection. 

### 2.6. Statistical Analysis

Descriptive analyses were performed with the built-in statistical functions in Microsoft Excel version 2016, whereas t-test calculations were performed with the R program (R version 4.0.2, 22 June 2020—“Taking off again”). Results are expressed as means ± SD.

## 3. Results

### 3.1. 8-OHdG and 8-Isoprostane Analytical Performance

Figure 1a,b give typical chromatograms of 8-isoprostane and 8-OHdG, respectively. These chromatograms were obtained after injection of 50 µL of a low standard level at 5 pg/mL, the lowest EBC quality check sample (5 pg/mL) and concentrated EBC sample collected from a healthy volunteer using the optimized conditions.

### 3.2. Sample Preparation

#### 3.2.1. Influence of Sample Concentration Process

We tested two concentrating methods based either on lyophilization or by using vacuum concentration. Figure 2 represents the mean slope of the linear regression based on five concentrated solutions for a minimum of two repetitions. We observed that glass containers needed a longer duration (about 9 h) to reach complete dryness compared to low-binding plastic tubes (about 6.5 h). 

We observed a significant and systematic decreased slope for 8-OHdG in glass vial (20–35%) and a statistically significant decreased slope (H_2_O lyoph glass *p* = 0.022; EBC lyoph glass *p* = 0.033; EBC lyoph plastic *p* = 0.041) for 8-isoprostane (35–40%) when the lyophilization was conducted at low pressure (0.06 mbar) compared to the standard without the concentration step. On the contrary, when using the vacuum concentration at a higher pressure (16 mbar), the slope was similar (8-isoprostane) or even significantly higher (8-OHdG) (*p* = 0.007) to that obtained without the concentration step. Nevertheless, this increase disappeared when the value was corrected with the SI concentration. Based on these results, we selected the vacuum concentration method for our sample preparations. 

#### 3.2.2. Effect of Protein Purification

Proteins are present in EBC in relatively high concentrations (typical range 0.76–107.7 µg/mL EBC [44]). We tested a clean-up procedure using a centrifugal concentrator system to remove high molecular weight proteins. This treatment had a strong impact on 8-OHdG with a two-fold decreasing signal (*p* = 0.001; *n* = 11) (Figure 3). Such a decrease can be attributed to the strong adsorption of 8-OHdG on the cellulose triacetate membrane [45]. 8-isoprostane, on the contrary, presented an increased signal of about 20% compared to the EBC sample without the concentration step. By washing the membrane with water before the sample treatment, we observed a decreased signal, similar to the one without centrifugation. This result suggests that an interfering compound soluble in water was present on the membrane, artificially increasing the signal of 8-isoprostane when the centrifugal concentrator system was used. Based on these results, we decided not to use such a clean-up procedure. 

Matrix effects were evaluated by comparing the slope of the calibration standards with the one obtained with the spiked EBCs at the same concentrations and without IS corrections using an unpaired t-test. No statistically significant difference was observed between the slopes from calibration standard not corrected by IS (5569 ± 684, 627 ± 220 for 8-OHdG and 8-isoprostane, respectively) and the EBC samples (4880 ± 661, 696 ± 1.53) for 8-OHdg and 8-isoprostane, respectively) (unpaired t-test, 8-OHdG *p* = 0.197; 8-isoprostane *p* = 0.831, *n* = 4).

#### 3.2.3. Effect of the Coating on Material Surface

Experiments of Tufvesson [12] showed that coating disposable polypropylene device with the detergent Tween 20 significantly increased the 8-isoprostane recovery. Nevertheless, we did not observe a difference on 8-OHdG and 8-isoprostane concentrations in our experiments comparing coated and not-coated systems (Appendix A). 

### 3.3. Method Validation

The characteristics of the method are given in Appendix A for 8-OHdG and Appendix A for 8-isoprostane. The LOD was 1 pg/mL EBC for 8-isoprostane and 0.5 pg/mL EBC for 8-OHdG. The recovery was between 90–110%. The repeatability for the two biomarkers was smaller than 20% for at the lowest concentration (5 pg/mL) and smaller than 6% for the higher concentration (15 pg/mL). The criteria for linearity were assessed by means of the coefficient of determination (R^2^), which was always above 0.99 in all analyses. 

### 3.4. Concentrations Measured in EBC Samples 

We did not detect 8-OHdG nor 8-isoprostane in EBC obtained from healthy volunteers in our laboratory or from the 26 workers (Appendix A). 

## 4. Discussion

We developed an LC–ESI–MS/MS analytical method that we thought to be sufficiently sensitive for quantifying low concentrations of 8-OHdG and 8-isoprostane concentrations in EBC. Although the validation steps resulted in an LOD < 1 pg/mL EBC for both oxidative stress biomarkers, they could not be detected in either the samples from healthy or asthmatic and COPD diseased workers. 

Based on Table 3 and Table 4, we observed that reported 8-isoprostane concentrations in EBC of healthy volunteers were quite variable and a function of the chemical analytical method used (LC–MS: 1–85 pg/mL EBC; GC–MS: 0.2–7 pg/mL EBC). For 8-OHdG (Table 4), the few reported concentrations in EBC are between 3–360 pg/mL EBC.

### 4.1. Method Optimization 

Our method development was based on [32,46], but we also wanted to simultaneously quantify 8-OHdG and 8-isoprostane for a fast and cost-effective analysis. This is challenging because the polarity and the chemical properties of the two biomarkers are quite different. 8-OHdG is the more polar of the two molecules due to the presence of polar functional groups (amides, hydroxyls, and amine), in contrast with 8-isoprostane, mostly formed by alkane chains. We abandoned a purification procedure to isolate potential interferences because this would further reduce the already low concentrations in either of the two compounds [61]. 

We decided to concentrate as much as possible on the EBC and considered two viable approaches: either lyophilization or centrifugation in conjunction with either low-binding plastics or glass. The concentrations obtained using lyophilization were much lower (20–40%) compared with the centrifugation approach while the evaporation times were similar, and this demonstrated a possible evaporation of the biomarkers at very low pressures. Consequently, we recommend the centrifugation approach and low-binding plastic materials when concentrating EBC samples to reduce 8-OHdG and 8-isoprostane losses. 

We were able to achieve sufficiently low LODs (<1 pg/mL) with this novel method to quantify 8-isoprostane and 8-OHdG according to reported baseline values [22,23]. This LOD was similar to five studies and lower than two studies (Table 3 and Table 4). We therefore expected our healthy and diseased workers to have isoprostane values comprised between LOD at 4 pg/mL and about 60 pg/mL, respectively (Table 3). For 8-OHdG, we expected healthy workers to be below 20 pg/mL and diseased workers around 36 pg/mL (Table 4). As we could not detect both biomarkers in any samples, we believe other factors hinder the quantification of these biomarkers in EBC.

### 4.2. Possible Reasons for Non-Detection of 8-OHdG and 8-Isoprostane in this Study

Many factors, in addition to sample concentration, can be responsible for the absence of detection and include collection time, storage of the EBC samples, type of EBC collection device and analytical issues. These parameters are considered here.

#### 4.2.1. Storage and Collection Time

To prevent artifactual changes in concentrations of biological markers after sampling, the samples were directly kept at −20 °C on field than transferred to −80 °C at the end of each collection day. Storage temperature of −80 °C is considered to provide the best storage temperature [62]. Because EBC samples could not be rapidly analyzed at the place of collection and laboratories are generally distant from sampling places, storage conditions could influence the analyte levels [63]. If we assume the same decay rate as reported by Syslova for our samples, stored in low-binding plastic tubes for 8 months, our results would have underestimated the concentrations of 8-isoprostane and 8-OHdG up to 14% depending on the amount of EBC obtained from the workers. Similarly, Havet et al. [63] measured exhaled 8-isoprostane concentrations 5–9 years after collection and only 30% of measurements were below the LOD. 8-OHdG is reported to be stable in water for *several* months at 4 °C [64] and at least two years when stored in acidic conditions such as in urine at −80 °C [65]. Given these results, we conclude that storage time does not represent a major concern for this study.

The EBC collection methodology has been suggested as one of many factors inhibiting satisfactory quantification of 8-isoprostane. Cooling temperature plays a critical role on the biomarker levels in exhaled breath condensate by influencing the condensation process on the device surface [5]. The TurboDECCS device that we are currently using is limited to −10 °C, so compounds could be viably altered if the cooling temperature during collection is insufficient. In one study, Goldoni et al. [66] showed a clear and significant trend toward increasing EBC volumes with decreasing collection temperature (0–> −10 °C), which affect both the concentration and absolute amounts of biomarkers. In the same way, Czebe et al. [67] observed that pH of EBC was influenced by the condensation temperature but not the protein concentration. In contrast, Zamuruyev et al. [68] have shown that the concentration of low-polarity non-volatile compounds (as is the case for 8-OHdG and 8-isoprostane) in EBC were practically independent of the collection temperature (tested between 0 °C and −56 °C). 

Several reports [10,12,22,46] also highlighted the possible influence of the condensing surface characteristics upon biomarker results. When sampling EBC, Rosias et al. [69] concluded that potential loss of biomarkers such as 8-isoprostane in the collection system could occur due to adsorption on the inner surface of the collection tubes. Some authors compared glass with a plastic-based collection device [69] or tried to passivate the surface of the condenser with different compounds (Tween-20, bovine serum albumin) [12,24]. Our results did not confirm these results, as we found no significant difference in 8-isoprostane and 8-OHdG concentrations when standards were in contact with Tween-20 treated or non-treated collection systems used with the TurboDeccs refrigeration device. This is consistent with results from Sood et al. [24]. We are thus confident that the undetected levels of 8-isoprostane and 8-OHdG in our EBC samples are not due to adsorption of the analytes on the surface of the used material.

#### 4.2.2. Protein Interferences 

A large amount of proteins can be found in EBC from healthy volunteers in a typical range of 0.76–107.7 µg/mL EBC [44]. These proteins are also concentrated in the EBC during the preparation of the samples and might suppress the quantification of the two biomarkers. Therefore, protein extraction might be necessary before the samples can be analyzed. Gonzalez-Reche et al. used an online LC column (LiChrospher^®^ ADS C18) to exclude macromolecules such as proteins (≥17 kDa) [70], but this did not significantly improve the quantification of 8-isoprostane in EBC obtained from healthy volunteers [47]. Ultrafiltration of EBC samples to reduce protein contamination using Vivaspin^®^ as described here, did not improve the LC–MS signal either. 

#### 4.2.3. Analytical Issues

All validation steps were performed following FDA/ICH guidelines [41]. Recovery experiments on EBC were performed to assess possible errors and losses in sensitivity arising from ion suppression. Excellent recoveries were obtained for 8-isoprostane (95–104%) and good recoveries were obtained for 8-OHdG (89–98%) as well as repeatability (<20% for lowest concentrations). The matrix effect was negligible when comparing slopes from calibration curves prepared in EBC and the standard sample, demonstrating an absence of potential interferences in the analysis as expected for EBC [70]. We do believe that the matrix effect is not the chemical analytical factor that hinders the detection of 8-OHdG and 8-isoprostane.

### 4.3. Variability in the Literature

Many other investigators have reported similar methodological problems. In the majority of the studies where 8-isoprostane was quantified in EBC, commercially available immunoassay kits were used. Even using immunoassay measurement, which is known to be prone to artefacts [71], few groups were unable to reliably measure 8-isoprostane concentrations in EBC samples [72,73,74]. Chemical analytical methods based on chromatography coupled to mass spectrometry detection are considered superior because of the enhanced sensitivity and selectivity over immunoassays. Nevertheless, even with such instrumental methods, results for 8-isoprostane and 8-OHdG in EBC are disparate as illustrated in Table 3 and Table 4 for 8-isoprostane and 8-OHdG, respectively. 

#### 4.3.1. 8-Isoprostane

Whereas Sanak et al. have successfully validated GC–MS methods for the determination of 8-isoprostane in EBC of healthy people [11], others, such as Carpenters and al. [10], have emphasized associated sensitivity problems. In their study, 30% of the control subjects exhibited 8-isoprostane concentrations below an excellent LOD (0.2 pg/mL). We also noted that GC–MS-based methods give EBC 8-isoprostane concentrations in healthy volunteers to be in the sub-pmol/mL range, and about 10 times lower compared with LC–MS-based methods (Table 3). Using LC–MS-based methods, a high heterogeneity in EBC levels is observed for healthy volunteers as presented in Table 3. Janicka et al. [46] reported 8-isoprostane concentrations in healthy volunteers below the LOD (1 pg/mL) using LC–MS/MS, suggesting that such levels of 8-isoprostane are very low. On the contrary, levels above 80 pg/mL have been reported for healthy volunteers [37].

Inclusion of purification procedures may improve the analytic performance by reducing endogenous contaminants or interferences present in EBC, thereby improving sensitivity. Syslova and colleagues [32] developed a method to separate 8-isoprostane from other isoprostane isomers using an immunosorbent affinity column. They were able to quantify 8-isoprostane in EBC from diseased and healthy subjects. Nevertheless, the reported 8-isoprostane concentrations in the Syslova et al. study for healthy volunteers were much higher compared to other studies (Table 3). Saliva contamination of EBC samples does not appear to be an issue in Syslova et al.’s study, as amylase activity in all samples did not exceed 0.1%. Nevertheless, Laumbach et al. [31] did not detect 8-isoprostane in the *majority* of their EBC samples, despite strictly adhering to Syslova et al.’s protocol. The same results were observed by Wang et al. [30] who also tried the Syslova et al. procedure and could not reproduce the results. Finally, by adding an online solid-phase extraction before LC–MS/MS, Wang et al. [30] managed to quantify 8-isoprostane in EBC of healthy subjects. Fritscher et al. [49] were able to quantify 8-isoprostane in EBC of healthy and diseases volunteers after adding a liquid–liquid extraction using ethyl acetate before LC–MS/MS analysis, while Gonzalez et al. [47] added an online extraction column (LiChrospher ADS C18 precolumn) but this failed to detect 8-isoprostane in EBC. It is not clear why 8-isoprostane was undetectable in some studies and detectable in the others. Horvath et al. [5] already reported that the measurement of 8-isoprostane was complicated due to the problems of reproducibility of assays by different groups with contradictory results. We did not observe any attenuated signal issues in our QC samples using a simple LC–MS/MS method. Furthermore, Liou [50], Wu [51] and Wang [30] were all part of the same laboratory, so the influence of purification on 8-isoprostane measurement could not be demonstrated. Alternatively, Battaglia et al. [75] found lower recovery rates (60%) of 8-isoprostane when they used an immunoaffinity sorbent. A possible explanation for this discrepancy could be that 8-isoprostane would conjugate with macro-molecules present in the EBC and remain undetectable [76]. The quantity of proteins can differ significantly between subjects [62] and this could explain the considerable variability in results. However, this problem of conjugation has never been reported in the scientific literature except for 8-isoprostane in plasma, where 8-isoprostane is bound to plasma lipids [77] and in urine, where 8-isoprostane is glucuronide conjugated and the amount conjugated vary between 30 and 80% of the total 8-isoprostane levels [76]. 

#### 4.3.2. 8-OHdG

Regarding 8-OHdG, its presence in EBC is still debatable [60,78], as a limited number of studies have reported concentrations in this matrix (Table 4) [23,55,59]. Only one team managed to quantify 8-OHdG by LC–MS/MS (Syslova and Pelclova being part of the same laboratory), which is considered as the gold standard method. Two other teams have analyzed it by immunoassays [23]. The fact that we failed to find 8-OHdG in any of our samples (<0.5 pg/mL EBC) suggests that this biomarker would be present at very low concentrations in the lung. This is in contradiction with data from the Syslova group [37], who reported concentrations between 12–19 pg/mL. Another hypothesis is that 8-OHdG would be rapidly adducted with lipids such as malondialdehyde present in airway lining fluid and would therefore be impossible to detect without prior separation [79].

Alternatively, 8-OHdG and 8-isoprostane previously reported in high concentrations in EBC may originate from sample contamination with saliva. In a review, Wang et al. [80] report concentrations of 8-OHdG and 8-isoprostane in the ng/mL range in the saliva of healthy volunteers. Among the commercial EBC collection devices, only the ECoScreen and TurboDeccs systems have built-in saliva traps, and these are the ones used by Pelclova and Syslova; but they may not be sufficiently effective [35]. Detecting salivary amylase is a frequently used method [5,32,81] to exclude saliva contamination in EBC. Nevertheless, α-amylase activity in the same way as any enzymatic method may not be sufficiently sensitive to show the presence of saliva in very small quantities [39]. Based on the Wang et al. levels reported in saliva [80], and hypothesizing a contribution of saliva in EBC of only 0.1% based on amylase analysis, 8-isoprostane levels in the range of some pg/mL could be expected. More sensitive or alternate LC–MS methods may be required to confirm saliva contamination [39]. 

Methods capable of reaching LOD under 0.1 pg/mL could perhaps help in determining 8-isoprostane and 8-OHdG in a reliable way. Presently, only GC–MS methods reach such low detection levels. One additional possibility to reach such low detection limits would be to collect larger EBC sample volumes and concentrate them as much as possible. However, this may not be feasible because it is hardly conceivable to increase the collection time, which normally lasts for 10–20 min [5], particularly for patients suffering from a respiratory pathology. Nevertheless, it is important to point out that even if it would be possible to quantify these two biomarkers by a more efficient method, the values would remain very low. The utility of EBC technique in detecting inflammation for research and clinical purposes may be compromised because it would depend on whether the differences could be found between patients and healthy subjects and whether these values would exceed the inter-individual variation within the two groups. Presently, it is still difficult to control for all variables that might affect 8-isoprostane or 8-OHdG concentrations, and consequently, comparisons of data obtained in different laboratories are difficult. There is a need to standardize the procedure for EBC collection and validate analytical chemical methods. Future directions include inter-laboratory tests with an identical protocol, which are essential to reach a consensus on the reference methods and then the normal-reference values for EBC biomarkers in the general population.

### 4.4. Recommendations

Considering our results and the literature data, we can make the following recommendations regarding the chemical analysis of 8-OHdG and 8-isoprostane in EBC: Use a sensitive alpha-amylase detection to quantify possible saliva contamination;Pre-concentrate the samples, drastically (by at least a factor of 10), prior to LC-MS analysis;Include a possible purification step prior to analysis;Control the adsorption phenomena on sampling and material surfaces;Have instruments capable of targeting LOD of the order of 0.1 pg/mL to expect to detect both components;Conduct inter-laboratory studies (round-robins);Standardize EBC collection devices for analysis of 8-OHdG and 8-isoprostane.

## 5. Conclusions

We developed a novel and robust method for simultaneous detection of 8-isoprostane and 8-OHdG in EBC; however, this method could not detect these biomarkers in EBC obtained from human subjects. The present study highlights difficulties in determining both oxidative stress biomarkers in EBC to distinguish health status. We suspect that the lack of detecting these biomarkers in our study is due to methodological issues in particular factors affecting EBC collections. We therefore recommend conducting inter-laboratory studies to standardize the chemical analytical methods as well as EBC collection devices for analysis of 8-OHdG and 8-isoprostane. 

## Figures and Tables

**Figure 1 antioxidants-11-00830-f001:**
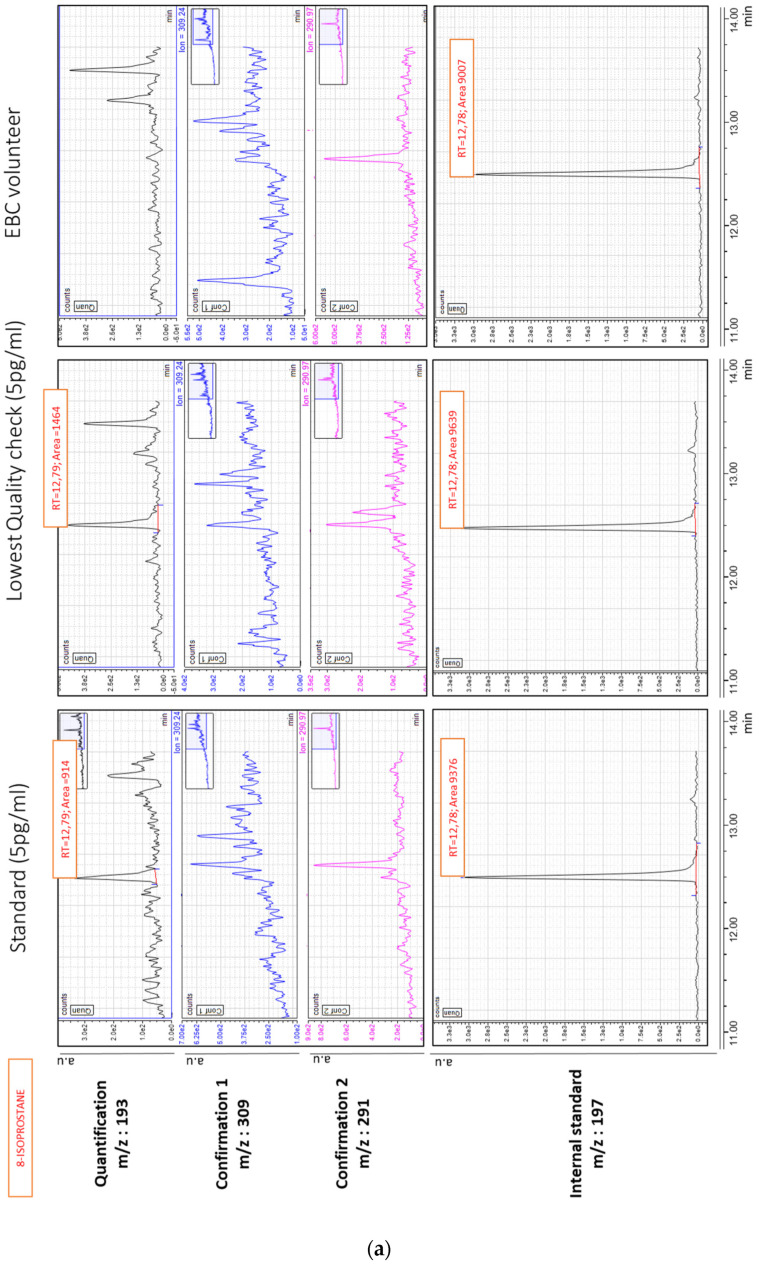
(**a**,**b**) Partial chromatogram of a standard injection at 5 pg/mL (left), of the lowest quality check (spiked EBC at 5 pg/mL, middle) and of a typical EBC sample (right), with retention times (RT) for 8-isoprostane and 8-OHdG, respectively. The units of the *y*-axis are in arbitrary units, counts per second. (**a**,**b**) are chromatograms obtained from the same healthy volunteer.

**Figure 2 antioxidants-11-00830-f002:**
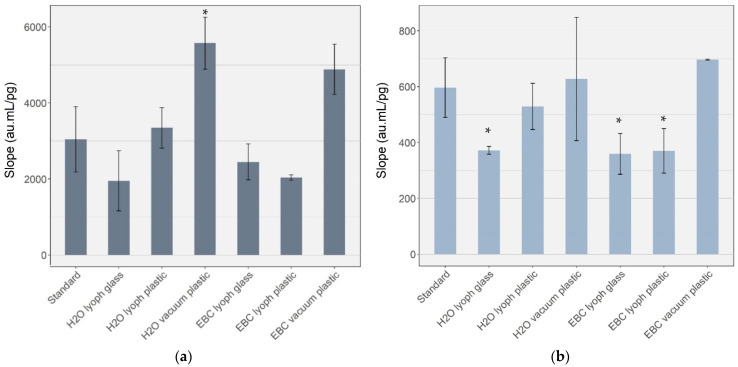
Slope signal (not corrected with IS) of the calibrations (minimum of 5 levels) obtained with water and EBC after lyophilization (lyoph) or vacuum concentration, in comparison with standard solutions without any concentration step for the biomarker (**a**) 8-OHdG and (**b**) 8-isoprostane, respectively. Error bars correspond to a minimum of two independent repetitions. au= Arbitrary unit; lyoph = lyophilization; standard = non-concentrated standard; H_2_O lyoph = concentrated standard. * Statistically significant difference (*p* < 0.05) compared to the standard without the concentration.

**Figure 3 antioxidants-11-00830-f003:**
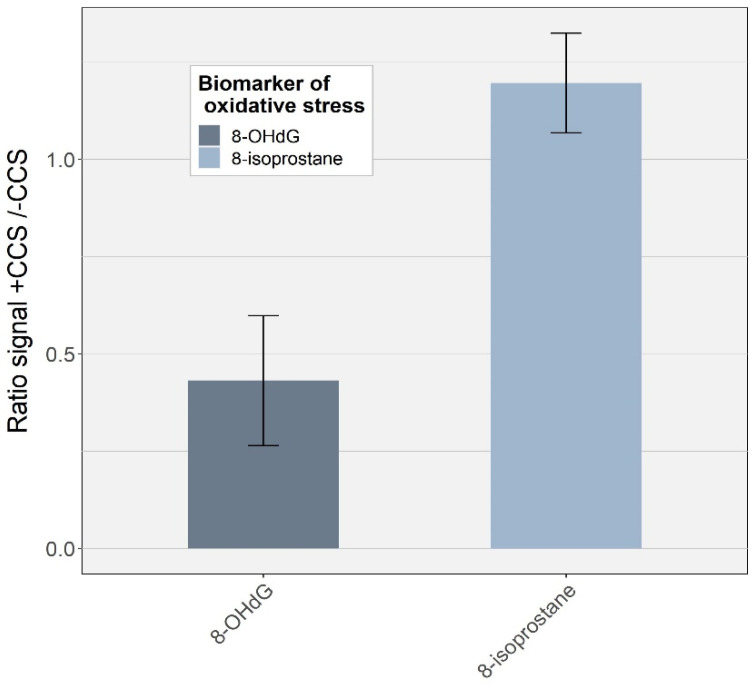
Effect of using the centrifugal concentrator system on the signal of both analytes in EBC. Bars correspond to standard deviation of three independent repetitions. CCS = centrifugal concentrator system.

**Table 1 antioxidants-11-00830-t001:** MS transitions and instrumental conditions for 8-OHdG and 8-isoprostane analysis. The vaporizer temperature was 350 °C with the ion transfer tube set at 390 °C. The argon gas pressure was set at 1.5 mTorr.

Compounds	Polarity	Mass Transitions (*m*/*z*)	Spray Voltage (V)	Collision Energy (V)	RF Lens (V)
8-OHdG	positive	284 → 140	3700	28.8	37
		**284 → 168**		10	37
		284 → 243		10.2	37
[^15^N5]-8-OHdG	positive	**289 → 173**	3700	10	40
8-isoprostane	negative	**353 → 193**	3400	25	80
		353 → 291		20	80
		353 → 309		20	80
8-isoprostane-d_4_	negative	**357 → 197**	3400	25	78

Mass transitions in bold are quantification transitions, others are confirmation transitions.

**Table 2 antioxidants-11-00830-t002:** Summary of participant demographics (*n* = 26).

	Healthy	Asthmatic	COPD
Number of subjects (male/female)	10 (6/4)	7 (3/4)	9 (6/3)
Age mean years ± SD (range)	51 ± 5.2 (44–60)	47 ± 5.2 (40–57)	54 ± 5.8 (41–60)
BMI mean kg/m^2^ ± SD (range)	25 ± 3.7 (20–32)	25 ± 2.4 (22–28)	24 ± 4.9 (19–34)
FEV1/FVC1 ratio	0.771 ± 0.04 (0.713–0.825)	0.665 ± 0.04 (0.597–0.697)	0.611 ± 0.06 (0.501–0.675)
Smokers (%)	30%	28%	66%

SD = standard deviation; BMI = body mass index; COPD = chronic obstructive pulmonary disease; FEV1 = forced expiratory volume in one second; FVC = forced vital capacity.

**Table 3 antioxidants-11-00830-t003:** Reported concentrations of 8-isoprostane in EBC in literature by validated RIA, GC–MS, and LC–MS methods.

Reference	Year	Study Group	Analytical Method	Collection Apparatus	Concentration of EBC	Lod	Basal Concentration
**Janicka** [46]	2012	healthy individuals	LC–MS/MS	TurboDeccs	lyophilized	1 pg/mL	**not detectable**
**Gonzalez** [47]	2009	healthy individuals	LC–MS/MS	Ecoscreen	-	5 pg/mL	**not detectable**
**Laumbach** [31]	2014	healthy individuals	affinity sorbant + LC–MS/MS	Ecoscreen	drying under nitrogen	2.5 pg/mL	**not detectable**
**Carpenter** [10]	1998	healthy individuals	GC–MS	Teflon-lined tubing (Tygon)	-	0.02 pg/mL *	**detectable in 3 of 10 control subjects (30%)** **7 ± 4 pg/mL ^e^**
**Sanak** [11]	2010	healthy individuals	GC–MS	Ecoscreen	drying under nitrogen	-	0.19 (0.14–0.29) pg/mL ^a^
**Sanak** [48]	2011	healthy individuals	GC–MS	Ecoscreen	drying under nitrogen	-	0.26 (0.2–0.47) pg/mL ^a^
**Fritscher** [49]	2012	healthy individuals	LC–MS/MS	RTube	drying under nitrogen	0.05–0.1 pg	0.9 (0.2–1.7) pg/mL ^d^
**Syslova** [37]	2010	healthy individuals	LC–MS/MS	Ecoscreen	lyophilized	8 pg/mL	86.7 (65.8–105.8) pg/mL ^a^
**Liou** [50]	2017	healthy individuals	affinity sorbant + LC–MS/MS	Ecoscreen	drying under nitrogen	1 pg/mL	3.14 (2.07) pg/mL ^c^
**Wu** [51]	2021	healthy individuals	affinity sorbant + LC–MS/MS	Ecoscreen	drying under nitrogen	1 pg/mL	3.930 (3.655) pg/mL ^c^
**Wang** [30]	2010	healthy individuals	affinity sorbant + LC–MS/MS	Ecoscreen	drying under nitrogen	1 pg/mL	4.44 ± 2.01 pg/mL ^b^
**Syslova** [32]	2008	healthy individuals	affinity sorbant + LC–MS/MS	Ecoscreen	drying under nitrogen	1 pg/mL	36 (20–55) pg/mL ^b^
**Santini** [52]	2016	healthy ex-smokers	RIA	Ecoscreen	-	2 pg/mL	8 (6.0–8.8) pg/mL ^e^
**Montuschi** [16]	2000	healthy individuals	RIA	Ecoscreen	-	4 pg/mL	10.8 ± 0.8 pg/mL ^b^
**Lucidi** [53]	2008	healthy individuals	RIA	Ecoscreen	-	10 pg/mL	15.5 (11.5–17.0) pg/mL ^d^
**Wu** [51]	2021	exposed people to carbon nanotubes	affinity sorbant + LC–MS/MS	Ecoscreen	drying under nitrogen	1 pg/mL	5.920 (9.040) pg/mL ^c^
**Liou** [50]	2017	exposed people to metal oxidenanoparticles	affinity sorbant + LC–MS/MS	Ecoscreen	drying under nitrogen	1 pg/mL	7.13 (8.21) pg/mL ^c^
**Syslova** [32]	2008	exposed people to asbestos	affinity sorbant + LC–MS/MS	Ecoscreen	drying under nitrogen	1 pg/mL	60 (50–70) pg/mL ^b^
**Santini** [52]	2016	smokers	RIA	Ecoscreen	-	2 pg/mL	11.2 (6.4–18.8) pg/mL ^e^
**Janicka** [46]	2012	smokers	LC–MS/MS	TurboDeccs	lyophilized	1 pg/mL	13–35 pg/mL
**Carpenter** [10]	1998	patients with ALI/ARDS	GC–MS	Teflon-lined tubing (Tygon)	-	0.02 pg/mL *	**detectable in 14 of 22 study patients (64%)** **87 ± 28 pg/mL ^e^**
**Mastalerz** [54]	2011	asthma patients	GC–MS	Ecoscreen	drying under nitrogen	-	0.25 ± 0.12 pg/mL ^b^
**Sanak** [48]	2011	asthma patients	GC–MS	Ecoscreen	drying under nitrogen	-	0.32 (0.15–0.3) pg/mL ^a^
**Corraro** [26]	2010	asthma patients	GC–MS	TurboDeccs	drying under nitrogen	3.9 pg/mL	68 (10.3) pg/mL ^e^
**Santini** [52]	2016	COPD patients	RIA	Ecoscreen	-	2 pg/mL	17.8 (8.8–31.2) pg/mL ^e^

ALI = and acute lung injury, ARDS = acute respiratory distress syndrome, RIA = radioimmunoassays. ^a^ Values are presented as medians (25th–75th percentiles); ^b^ Values are presented as mean (±SD); ^c^ Values are presented as medians (IQR); ^d^ Values are presented as median (range); ^e^ Values are presented as mean (±SEM); * LOD obtained from a publication of [30].

**Table 4 antioxidants-11-00830-t004:** Reported EBC concentrations of 8-OHdG in the literature by validated ELISA and LC–MS methods.

Reference	Year	Study Group	Analytical Method Approach	Collection Apparatus	Concentration of Ebc	Lod	Basal Concentration
**Fireman** [55]	2019	healthy individuals	ELISA kit	TurboDeccs	-	-	3 pg/mL
**Pelclova** [56]	2012	healthy individuals	LC–MS/MS	EcoScreen	-	7 pg/mL	10 (9.0–11.0) pg/mL ^c^
**Pelclova** [57]	2016	healthy individuals	LC–MS/MS	EcoScreen	-	7 pg/mL	13 (11.5–14.5) pg/mL ^c^
**Syslova** [37]	2010	healthy individuals	LC–MS/MS	EcoScreen	-	7 pg/mL	14.8 (12.8–19.9) pg/mL ^a^
**Pelclova** [58]	2018	healthy individuals	LC–MS/MS	EcoScreen	-	7 pg/mL	18 (15.0–21.0) pg/mL ^c^
**Doruk** [59]	2011	healthy individuals	ELISA kit	EcoScreen	-	41 pg/mL	360 ± 90 pg/mL ^b^
**Graczyk** [60]	2017	exposed welders	ELISA kit	R-tube		-	**not detectable**
**Doruk** [59]	2011	smokers	ELISA kit	EcoScreen	-	41 pg/mL	520 ± 150 pg/mL ^b^
**Doruk** [59]	2011	passive smokers	ELISA kit	EcoScreen	-	41 pg/mL	310 ± 100 pg/mL ^b^
**Fireman** [55]	2019	COPD patients	ELISA kit	TurboDeccs	-	-	36 pg/ml
**Syslova** [37]	2010	silica- or asbestos-disorders due to occupational exposure patients	LC–ESI–MS/MS	EcoScreen	-	7 pg/mL	46.5 (39.4–49.9) pg/mL ^a^

^a^ Values are presented as medians (25th–75th percentiles); ^b^ Values are presented as mean (±SD); ^c^ Values are presented as median (range).

## Data Availability

All of the data is contained within the article and the Appendix A.

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
