# Peer review of "Challenges in Quantifying 8-OHdG and 8-Isoprostane in Exhaled Breath Condensate"

_antioxidants, 2022, doi:10.3390/antiox11050830_

Round 1
Reviewer 1 Report
The study is encouraging given that non-invasive alternatives are being sought to monitor so-called dysfunction. The authors present through this study an intense work on the detection of two expiratory respiration biomarkers. They suggest that 8-hydroxydeoxyguanosine (8-OHdG) and 8-isoprostane in EBC can be considered as potential candidates for respiratory biomarkers in lung diseases associated with inflammation and oxidative stress.
This manuscript is good in the quality of writing, the section Materials and Methods and Method validation and applicability are very well described. Also, the authors provide sufficient information regarding the study population.
Despite all the work done, the entire manuscript focuses only on the method, calibration and preparation of samples, while the results section and the discussion section represent a very small percentage of the entire manuscript.
Suggestion:
- Is term B missing from equation [Analyte]EBC = (Ssple/A)*(1/13.33)?
- The sample preparation method is very complex. Exists the possibility to alter the EBC sample?
- The method proposed is very complex and what's the difference or advantages of the proposed method compared with others conventional breath diagnosis?
In addition, I am not convinced that it could be published in Antioxidants.

Author Response
The study is encouraging given that non-invasive alternatives are being sought to monitor so-called dysfunction. The authors present through this study an intense work on the detection of two expiratory respiration biomarkers. They suggest that 8-hydroxydeoxyguanosine (8-OHdG) and 8-isoprostane in EBC can be considered as potential candidates for respiratory biomarkers in lung diseases associated with inflammation and oxidative stress.
This manuscript is good in the quality of writing, the section Materials and Methods and Method validation and applicability are very well described. Also, the authors provide sufficient information regarding the study population.
Authors’ response: Thank you very much for this positive appreciation.
Despite all the work done, the entire manuscript focuses only on the method, calibration and preparation of samples, while the results section and the discussion section represent a very small percentage of the entire manuscript.
Authors’ response: This manuscript is primarily a methodological article, whose aim was to develop an analytical method allowing the simultaneous determination of two biomarkers of oxidative stress in EBC. As we could not detect nor quantify these biomarkers in EBC as expected, we wanted to understand the reasons for this. Such process require precise description and validation of the method and assessment of possible factors able or thought to interfere with the quantification. That is the reason why a large focus is devoted to the description/validation of the method in the results section, whereas in the discussion section, we try to identify the reasons possibly explaining the absence of detection and the large discrepancies observed in the literature.
We have modified the titles in the manuscript to make them more explicit and to facilitate the reading of the manuscript.
Suggestion:
- Is term B missing from equation [Analyte]EBC = (Ssple/A)*(1/13.33)?
Authors’ response: We thank the reviewer for detecting this error. The term B has been added to the equation.
- The sample preparation method is very complex. Exists the possibility to alter the EBC sample?
Authors’ response: As explained in the objective of the manuscript, we attempted to make the sample preparation and analysis as simple as possible. We developed a simple and robust preparation and analysis method because it was planned for routine screening of populations where a cost-effective analysis is needed. We did not use a pre-column or purification step. We just concentrated EBC to enable us to detect our compounds of interest in picomolar range (see table 3 and 4). Finally, we simply reconstituted the residue and injected it directly into the apparatus.
We added an antioxidant (butylated hydroxytoluene, BHT) in the preparation of the samples in order to prevent the occurrence of auto-oxidation that could lead to the production of ex-vivo 8-isoprostane (1, 2) or without impact on 8-OHdG (3). We added this detail in lines 169 and 170.
During our tests, we did not observe any effect of BHT on 8-OHdG or 8-isoprostane concentration in EBC. In a similar way, we did not find any evidence in the scientific literature that BHT could have negative effect on the level of biomarkers in EBC.
(1) Milatovic, Dejan, and Michael Aschner. “Measurement of isoprostanes as markers of oxidative stress in neuronal tissue.” Current protocols in toxicology vol. Chapter 12,Supplement 39 (2009): Unit12.14. doi:10.1002/0471140856.tx1214s39
(2) Morrow JD, Roberts LJ 2nd. Mass spectrometric quantification of F2-isoprostanes in biological fluids and tissues as measure of oxidant stress. Methods Enzymol. 1999;300:3-12. doi: 10.1016/s0076-6879(99)00106-8. PMID: 9919502.
(3) Syslová K, Kačer P, Kuzma M, Pankrácová A, Fenclová Z, Vlčková S, Lebedová J, Pelclová D. LC-ESI-MS/MS method for oxidative stress multimarker screening in the exhaled breath condensate of asbestosis/silicosis patients. J Breath Res. 2010 Mar;4(1):017104. doi: 10.1088/1752-7155/4/1/017104. Epub 2010 Jan 7. PMID: 21386209.
- The method proposed is very complex and what's the difference or advantages of the proposed method compared with others conventional breath diagnosis?
Authors’ response:
Conventional methods for breath analysis do not analyse specific biomarkers but rather reflect global ongoing metabolic internal process by the measure of a pattern of volatile compounds (4).
8-isoprostane and 8-OHdG are among the most widely used non-volatile biomarkers of oxidative stress. Their concentration in exhaled breath condensate can be determined by several methods, and the most commonly used is the enzyme-linked immunosorbent assay (ELISA). ELISA is a simple, fast and relatively inexpensive method. Nevertheless, we cannot use this method because of interferences and possible cross-reactivity between these biomarkers and structurally similar isomers. In comparison, the use of LC-MS/MS allows us to quantify adequately and accurately the biomarkers in EBC. The difference and advantages of the proposed method compared to ELISA are already indicated in the introduction, so no changes in the manuscript have been made.
Compared to other protocols already published, our method for quantifying these effect biomarkers in EBC using a mass spectrometry method is quite simple. For example, the use of GC-MS/MS method as described by Sanak et al. (5) requires an extensive manual sample preparation with a complex two-step derivatization process. Concerning LC-MS method, Gonzalez-Reche et al. (6), Wang et al.(7) have to use another LC column to exclude macromolecules such as proteins. Fritscher et al. (8) add a liquid-liquid extraction using ethyl acetate before LC-MS/MS analysis. Finally, Syslova et al., (9) Laumbach et al. (10) and Wu et al.(11) include an expensive and complex purification procedure based on immuno-affinity column to reduce endogenous contaminants present in EBC. Nothing has been changed in the manuscript concerning this last aspect.
(4) Gaugg MT et al. Real-Time Breath Analysis Reveals Specific Metabolic Signatures of COPD Exacerbations. Chest. 2019 Aug;156(2):269-276. doi: 10.1016/j.chest.2018.12.023. Epub 2019 Jan 24. PMID: 30685334.
(5) Sanak, M.; Gielicz, A.; Nagraba, K.; Kaszuba, M.; Kumik, J.; Szczeklik, A. Targeted eicosanoids lipidomics of exhaled breath condensate in healthy subjects. J Chromatogr B Analyt Technol Biomed Life Sci 2010, 878, 1796-1800, doi:10.1016/j.jchromb.2010.05.012.
(6) Gonzalez-Reche, L.M.; Musiol, A.K.; Müller-Lux, A.; Kraus, T.; Göen, T. Method optimization and validation for the simultaneous determination of arachidonic acid metabolites in exhaled breath condensate by liquid chromatography-electrospray ionization tandem mass spectrometry. J Occup Med Toxicol 2006, 1, 5, doi:10.1186/1745-6673-1-5.
(7) Wang, C.J.; Yang, N.H.; Liou, S.H.; Lee, H.L. Fast quantification of the exhaled breath condensate of oxidative stress 8-iso-prostaglandin F2alpha using on-line solid-phase extraction coupled with liquid chromatography/electrospray ionization mass spectrometry. Talanta 2010, 82, 1434-1438, doi:10.1016/j.talanta.2010.07.015.
(8) Fritscher, L.G.; Post, M.; Rodrigues, M.T.; Silverman, F.; Balter, M.; Chapman, K.R.; Zamel, N. Profile of eicosanoids in breath condensate in asthma and COPD. Journal of Breath Research 2012, 6, 026001, doi:10.1088/1752-7155/6/2/026001.
(9) Syslová, K.; Kacer, P.; Kuzma, M.; Klusâ ková, P.; Fenclová, Z.; Lebedová, J.; Pelclova, D. Determination of 8-iso-prostaglandin F(2alpha) in exhaled breath condensate using combination of immunoseparation and LC-ESI-MS/MS. Journal of chromatography. B, Analytical technologies in the biomedical and life sciences 2008, 867 1, 8-14.
(10) Laumbach, R.J.; Kipen, H.M.; Ko, S.; Kelly-McNeil, K.; Cepeda, C.; Pettit, A.; Ohman-Strickland, P.; Zhang, L.; Zhang, J.; Gong, J.; et al. A controlled trial of acute effects of human exposure to traffic particles on pulmonary oxidative stress and heart rate variability. Part Fibre Toxicol 2014, 11, 45, doi:10.1186/s12989-014-0045-5.
(11) Wu, W.-T.; Jung, W.-T.; Lee, H.-L. Lipid peroxidation metabolites associated with biomarkers of inflammation and oxidation stress in workers handling carbon nanotubes and metal oxide nanoparticles. Nanotoxicology 2021, 15, 577-587, doi:10.1080/17435390.2021.1879303.
- In addition, I am not convinced that it could be published in Antioxidants.
Authors’ response: We believe that this manuscript fits well into the scope of the journal because it is interested in development of innovative techniques and new methodologies for the analysis of the different components of oxidative stress involved in physio-pathological mechanisms of many diseases.
A part of the special Issue “The 10th Anniversary of Antioxidants: Past, Present and Future”, this article aims to summarize the current knowledge while illustrating future challenges for the characterization of trace biomarkers such as 8-isoprostane and 8-OHdG in highly diluted matrices like EBC.

Reviewer 2 Report
The authors attempted to quantify 8-OHdG and 8-iso in exhaled breath condensates (EBC) from healthy volunteers or patients with asthma or COPD. While the authors have claimed that their method is novel and robust, they were unable to detect both 8-OHdG and 8-iso in the EBC.
Comments:
- The title should be changed to indicate that this manuscript failed to detect 8-OHdG and 8-iso in EBC.
- Did the authors measure levels of 8-OHdG and 8-iso in patients to check if they are confounders? (i.e.: the patients do express higher levels perhaps in the blood, but it was not detectable in EBC)
- The authors mentioned workers – were the human volunteers obtained from the same workplace?
- On a similar question pertaining samples collection, were they collected in the lab/hospital and then readily stored in -80oC, or was there some delay between sample collection and storing in the freezer?
- The authors suggested that future technique(s) with sensitivity of 0.1 pg/ml – is that enough and what was the detection limit of the current technique the authors had employed?
- Would it affect the compounds viability if EBC was collected using a device at -20oC?
Author Response
Response to reviewer #2’s comments:
The authors attempted to quantify 8-OHdG and 8-iso in exhaled breath condensates (EBC) from healthy volunteers or patients with asthma or COPD. While the authors have claimed that their method is novel and robust, they were unable to detect both 8-OHdG and 8-iso in the EBC.
Comments:
The title should be changed to indicate that this manuscript failed to detect 8-OHdG and 8-iso in EBC.
Authors’ response: Whereas the analytical method itself could be validated, we have not been able to quantify 8-OHdG and 8-isoprostane in EBC samples of healthy or sick people. According to the literature, our method would be sensitive enough to allow a robust quantification of these biomarkers, but we were surprised not to succeed in this endeavor. Our results suggest that the very variable literature results about 8-OHdG and 8-isoprostane have to be considered with caution.
The title has been rephrased accordingly.
Did the authors measure levels of 8-OHdG and 8-iso in patients to check if they are confounders? (i.e.: the patients do express higher levels perhaps in the blood, but it was not detectable in EBC)
Authors’ response: We assumed that patients with asthma or COPD had a higher concentration of biomarkers in the EBC (1,2,3). Unfortunately, we did not see any difference between our volunteers (table S4). One explanation could be that our patients were not in exacerbate /inflammatory state during the collection period.
We did not collect blood because it is an invasive method and therefore undesirable for routine screening of the population.
(1) Fireman Klein, E.; Adir, Y.; Krencel, A.; Peri, R.; Vasserman, B.; Fireman, E.; Kessel, A. Ultrafine particles in airways: a novel marker of COPD exacerbation risk and inflammatory status. Int J Chron Obstruct Pulmon Dis 2019, 14, 557-564, doi:10.2147/COPD.S187560.
(2) Montuschi, P.; Collins, J.V.; Ciabattoni, G.; Lazzeri, N.; Corradi, M.; Kharitonov, S.A.; Barnes, P.J. Exhaled 8-isoprostane as an in vivo biomarker of lung oxidative stress in patients with COPD and healthy smokers. Am J Respir Crit Care Med 2000, 162, 1175-1177, doi:10.1164/ajrccm.162.3.2001063.
(3) Montuschi, P.; Corradi, M.; Ciabattoni, G.; Nightingale, J.; Kharitonov, S.A.; Barnes, P.J. Increased 8-isoprostane, a marker of oxidative stress, in exhaled condensate of asthma patients. Am J Respir Crit Care Med 1999, 160, 216-220, doi:10.1164/ajrccm.160.1.9809140.
The authors mentioned workers – were the human volunteers obtained from the same workplace?
Authors’ response: The human volunteers were obtained from the same workplace. We added this specification in the text (§ 2.5.2).
On a similar question pertaining samples collection, were they collected in the lab/hospital and then readily stored in -80oC, or was there some delay between sample collection and storing in the freezer?
Authors’ response: The exhaled breath condensate was collected and immediately frozen at -20°C in the field then stored at - 80°C at the end of each collection day. Biological sample collection, aliquoting and storage were operated in a closed clean room in RATP's medical department. Two people were working in parallel on this task, so no delays between sample collection and storing was encountered. The samples were then transported from Paris to Lausanne at a maximum temperature of -10°C for one day (controlled with an electric thermometer) and then stored at - 80°C in our laboratory until analysis. We have summarized the additional information in the § 2.3. and 4.2.1.
The authors suggested that future technique(s) with sensitivity of 0.1 pg/ml – is that enough and what was the detection limit of the current technique the authors had employed?
Authors’ response: As mentioned in § 3.3, the current LOD is 1 pg/ml for 8-isoprostane and 0.5 pg/ml for 8-OHdG. If we refer to Table 3 and in particular, to the GC-MS results, we observe that a LOD of 0.1 pg/ml (as reported by Sanak et al.) could be enough to quantify 8-isoprostane. This is the reason why we recommend having chemical analytical instruments capable of targeting a LOD 10 times lower than the one we found, if we expect to detect these two biomarkers. This part is mentioned in the discussion (line 619), so nothing has been changed in the manuscript.
Would it affect the compounds viability if EBC was collected using a device at -20oC?
Authors’ response: The TurboDECCS device that we are currently using is unable to go below -10°C. Nevertheless, it is known that cooling temperature play a critical role on the biomarker levels in exhaled breath condensate by influencing the condensation process on the device surface. In one study, Goldoni et al. (1) showed a clear and significant trend toward increasing EBC volumes with decreasing collection temperature (0°C -> -10°C), which affect both the concentration and absolute amounts of biomarkers. In the same way, Czebe et al. (2) observed that pH of EBC was influenced by the condensation temperature but not the protein concentration. Additionally, Zamuruyev (3) have shown that the concentration of low-polarity non-volatile compounds (as it is the case for 8-OHdG and 8-isoprostane) in EBC were practically independent of the collection temperature (tested between 0°C and -56°C).
We added this assertion and the corresponding new references in the text (§ 4.2.1).
(1) Goldoni, M., Caglieri, A., Andreoli, R. et al. Influence of condensation temperature on selected exhaled breath parameters. BMC Pulm Med 5, 10 (2005). https://doi.org/10.1186/1471-2466-5-10
(2) Krisztina Czebe, Imre Barta, Balázs Antus, Márta Valyon, Ildikó Horváth, Tamás Kullmann, Influence of condensing equipment and temperature on exhaled breath condensate pH, total protein and leukotriene concentrations, Respiratory Medicine, Volume 102, Issue 5, 2008, Pages 720-725, https://doi.org/10.1016/j.rmed.2007.12.01
(3) Zamuruyev, K. O., Borras, E., Pettit, D. R., Aksenov, A. A., Simmons, J. D., Weimer, B. C., Schivo, M., Kenyon, N. J., Delplanque, J. P., & Davis, C. E. (2018). Effect of temperature control on the metabolite content in exhaled breath condensate. Analytica chimica acta, 1006, 49–60. https://doi.org/10.1016/j.aca.2017.12.025

Reviewer 3 Report
This negative study is very important. It is honest and should be published.
The are minor issues to be solved:
1. good sensitivity must be defined - appearantly, the method is not good enough
2. the reasons for the differences to published positive outcomes have to be clear even from the abstract
3. the conclusion is weak - it is not true that a comparison is not possible, on contrary, it is hihgly needed, but with regards to the methodological issues.
4. Loss of analytes could be a reason, but it is not clear from the obtained results.
5. Even if all analytical problems are resolved, the specificity of a DNA damage marker is questionable.
6. Such a study requires a detailed systematic review of the published methods, concentrations, sensitivities... that is covered by the Tables, but the Discussion includes suggestions that are not based on the data obtained or on the comparison with the published studies. This is to be done.
7. Have the authors tried to analyze other biological samples? urine? plasma? saliva? tissue homogenates?
8. The graphs from Excel could be polished a bit...
Author Response
Response to reviewer #3’s comments:
This negative study is very important. It is honest and should be published.
Authors’ response: We thank the reviewer for the positive comments and for highlighting the importance of the method.
The are minor issues to be solved:
- good sensitivity must be defined - apparently, the method is not good enough
Authors’ response: We defined the good sensitivity as the level of concentration necessary for the detection of these biomarkers i.e. “picomolar range”. We added this term in the abstract. The current detection limits of our method mentioned in the abstract is 1 pg/ml for 8-isoprostane and 0.5 pg/ml for 8-OHdG. Knowing that molecular weight of 8-isoprostane is 354 g/mol and 8-OHdG is 299.24 g/mol, these value lies in the picomolar range. This range is good enough as it is quite similar with what is observed in the literature for LC-MS/MS (Table 3-4).
Calculation: 8-isoprostane 1/354.5= 0.0028 pmol/mL or 2.8pmol/L; 8-OHdG 0.5/299.24= 0.0016 pmol/mL or 1.6 pmol/L
- the reasons for the differences to published positive outcomes have to be clear even from the abstract
Authors’ response: The analytical method itself is validated. By deduction, the factors that could influence the results could be derived from the collection of EBC. We added a sentence in the abstract to indicate that the collection could be the main factor at the origin of the absence of detection or the possible overestimation of the values reported in the scientific literature.
- The conclusion is weak - it is not true that a comparison is not possible, on contrary, it is highly needed, but with regard to the methodological issues.
Authors’ response: The manuscript text (abstract and conclusion) has been rephrased as suggested by the reviewer.
- Loss of analytes could be a reason, but it is not clear from the obtained results.
Authors’ response: We tested the possible loss of analytes by surface adsorption as described in the § 3.2.3. We added a figure in supplementary material to show the obtained results.
- Even if all analytical problems are resolved, the specificity of a DNA damage marker is questionable.
Authors’ response: Oxidative DNA damage can be caused by chronic inflammation or induce production of ROS by exogenous factors (exposure to particulate matter for example) (4,5). Consequently, quantifying ROS in EBC could reflect the level of oxidative stress in the lung and be predictive of the development of respiratory diseases. Of course, the specificity of 8-OHdG is questionable. It is true that DNA damage markers are not disease-specific and can be influenced by other individual, behavioural or environmental factors. Nevertheless, by combining the measure of 8-OHdG with a panel of other oxidative stress biomarkers such as 8-isoprostane and MDA (6), we could constitute a characteristic and specific "fingerprint" which would be useful to diagnose a specific respiratory disease.
(4) Loft, S.; Olsen, A.; Møller, P.; Poulsen, H.E.; Tjønneland, A. Association between 8-oxo-7,8-dihydro-20-deoxyguanosine Excretion and Risk of Postmenopausal Breast Cancer: Nested Case-Control Study. Cancer Epidemiol. Biomark. Prev. 2013, 22, 1289–1296.
(5) Loft, S.; Svoboda, P.; Kawai, K.; Kasai, H.; Sørensen, M.; Tjønneland, A.; Vogel, U.; Møller, P.; Overvad, K.; Raaschou-Nielsen, O. Association between 8-oxo-7,8-dihydroguanine excretion and risk of lung cancer in a prospective study. Free Radic. Biol. Med. 2012, 52, 167–172.
(6) Hemmendinger M, Sauvain JJ, Hopf NB, Wild P, Suárez G, Guseva Canu I. Method Validation and Characterization of the Associated Uncertainty for Malondialdehyde Quantification in Exhaled Breath Condensate. Antioxidants (Basel). 2021 Oct 22;10(11):1661. doi: 10.3390/antiox10111661. PMID: 34829532; PMCID: PMC8615247.
- Such a study requires a detailed systematic review of the published methods, concentrations, sensitivities... that is covered by the Tables, but the Discussion includes suggestions that are not based on the data obtained or on the comparison with the published studies. This is to be done.
Authors’ response: We attempted to understand why we could not detect 8-OHdG and 8-isoprostane in EBC whereas the scientific literature mentions relatively high levels of these two biomarkers in this matrix. This is a difficult process and we can only suggest hypotheses at this point. This paper does not have the ambition to address all these hypotheses but to highlight some recommendations in order to improve the analysis of these biomarkers in the future. The recommendations we have made are all based either on our own results or on literature (Tables 3 and 4).
- High levels of 8-OHdG and 8-isoprostane in some publications might be due to a non-negligible salivary contamination (§ 4.3.2.).
- Pre-concentration is essential because the matrix is highly diluted. The factor 10 corresponds more or less to the factor used in our method (x13.33). Assuming that the level of biomarkers is in the range of 0.2-0.3 pg/ml (Table 3-4), this concentration factor would allow to detect these biomarkers in the EBC with the current sensitivity of LC-MS technique.
- The purification recommendation is based on the literature (it is mentioned in the table 3 as "affinity sorbent" in the analytical method column). In our study the removal of proteins, using ultrafiltration with Vivaspin® did not bring any improvement (§ 3.2.2.).
- Surface adsorption is often mentioned as an interfering factor in the literature. In our study and with our conditions (0.01% Tween 20, TurboDECCS device), the surface adsorption could not explain the lack of detection (Supplementary material, figure S1).
- The detection limit of 0.1 pg/ml for 8-isoprostane in EBC is based on the results of the literature and corresponds to the measurements made by GC-MS (Table 3).
- Interlaboratory testing will therefore help to standardize the methods to reach a consensus in the field as recommended by Horvath et al. (7).
The previous recommendations are already mentioned in the discussion so no changes have been made in the manuscript.
(7) Horváth I, Barnes PJ, Loukides S, Sterk PJ, Högman M, Olin AC, Amann A, Antus B, Baraldi E, Bikov A, Boots AW, Bos LD, Brinkman P, Bucca C, Carpagnano GE, Corradi M, Cristescu S, de Jongste JC, Dinh-Xuan AT, Dompeling E, Fens N, Fowler S, Hohlfeld JM, Holz O, Jöbsis Q, Van De Kant K, Knobel HH, Kostikas K, Lehtimäki L, Lundberg J, Montuschi P, Van Muylem A, Pennazza G, Reinhold P, Ricciardolo FLM, Rosias P, Santonico M, van der Schee MP, van Schooten FJ, Spanevello A, Tonia T, Vink TJ. A European Respiratory Society technical standard: exhaled biomarkers in lung disease. Eur Respir J. 2017 Apr 26;49(4):1600965. doi: 10.1183/13993003.00965-2016. PMID: 28446552.
- Have the authors tried to analyze other biological samples? urine? plasma? saliva? tissue homogenates?
Authors’ response: In parallel, we have developed a method (8) allowing the reliable measure of these two biomarkers in urine of volunteers. These results will be valorized in a future publication.
(8) Sambiagio, N., Sauvain, J. J., Berthet, A., Auer, R., Schoeni, A., & Hopf, N. B. (2020). Rapid Liquid Chromatography-Tandem Mass Spectrometry Analysis of Two Urinary Oxidative Stress Biomarkers: 8-oxodG and 8-isoprostane. Antioxidants (Basel, Switzerland), 10(1), 38. https://doi.org/10.3390/antiox10010038
- The graphs from Excel could be polished a bit...
Authors’ response: As suggested by the reviewer we polished all the graphs using R/ggplot 2 instead of Excel.

Round 2
Reviewer 1 Report
Throughout this research, the authors attempted to quantify 8-OHdG and 8-iso in exhaled breath condensates (EBC) from healthy volunteers or patients with asthma or COPD, as candidate biomarkers of lung diseases associated with inflammation and oxidative stress.
While they present a higher sensitivity of the liquid chromatography coupled to the tandem mass spectrometry method, they were unable to detect both 8-OHdG and 8-iso in the EBC.
However, the authors conducted intensive and complex research for this study. Even if the authors had a negative result in quantifying 8-OHdG and 8-isoprostane biomarkers, I believe that this research is important and should be published.

Author Response
Throughout this research, the authors attempted to quantify 8-OHdG and 8-iso in exhaled breath condensates (EBC) from healthy volunteers or patients with asthma or COPD, as candidate biomarkers of lung diseases associated with inflammation and oxidative stress.
While they present a higher sensitivity of the liquid chromatography coupled to the tandem mass spectrometry method, they were unable to detect both 8-OHdG and 8-iso in the EBC.
However, the authors conducted intensive and complex research for this study. Even if the authors had a negative result in quantifying 8-OHdG and 8-isoprostane biomarkers, I believe that this research is important and should be published.
Authors’ response: We thank the reviewer for the positive comments and for highlighting the importance of this research on 8-OHdG and 8-isoprostane in exhaled breath condensate.
Reviewer 2 Report
The authors had answered most of my previous comments satisfactorily, with the exception for two of them.
1)
"The authors suggested that future technique(s) with sensitivity of 0.1 pg/ml – is that enough and what was the detection limit of the current technique the authors had employed?
Authors’ response: As mentioned in § 3.3, the current LOD is 1 pg/ml for 8-isoprostane and 0.5 pg/ml for 8-OHdG. If we refer to Table 3 and in particular, to the GC-MS results, we observe that a LOD of 0.1 pg/ml (as reported by Sanak et al.) could be enough to quantify 8-isoprostane. This is the reason why we recommend having chemical analytical instruments capable of targeting a LOD 10 times lower than the one we found, if we expect to detect these two biomarkers. This part is mentioned in the discussion (line 619), so nothing has been changed in the manuscript."
^ I understand that moving the detection limit down from 1pg/ml to 0.1 pg/ml is increasing the sensitivity by 10x, however, this is assuming the samples are in the range of 0.1 - 0.99 pg for example. Perhaps this might help the detection of both mediators in patients with asthma and/or COPD during their inflamed status, but it is unlikely to be detected in healthy volunteers.
2)
"Did the authors measure levels of 8-OHdG and 8-iso in patients to check if they are confounders? (i.e.: the patients do express higher levels perhaps in the blood, but it was not detectable in EBC)
Authors’ response: We assumed that patients with asthma or COPD had a higher concentration of biomarkers in the EBC (1,2,3). Unfortunately, we did not see any difference between our volunteers (table S4). One explanation could be that our patients were not in exacerbate /inflammatory state during the collection period.
We did not collect blood because it is an invasive method and therefore undesirable for routine screening of the population."
^ Since the volunteers were obtained from the same workplace, there is a small chance the samples obtained here do not follow normal distribution or perhaps are somewhat skewed in this case.
The ability to determine if these patients express higher levels of 8-isoprostane and/or 8-OHdG systematically or in the lungs, but just not present in sufficient quantities in EBC.
If patients with asthma and/or COPD require exacerbation state in order to detect 8-isoprostane and 8-OHdG, then the detection for such a parameter would be somewhat redundant since spirometry would probably provide a better measurement in real-time, especially during exacerbation. On that note, what is the scientific rationale to measure both mediators if they are present only during exacerbation state?
Author Response
The authors had answered most of my previous comments satisfactorily, with the exception for two of them.
1)"The authors suggested that future technique(s) with sensitivity of 0.1 pg/ml – is that enough and what was the detection limit of the current technique the authors had employed?
Authors’ response: As mentioned in § 3.3, the current LOD is 1 pg/ml for 8-isoprostane and 0.5 pg/ml for 8-OHdG. If we refer to Table 3 and in particular, to the GC-MS results, we observe that a LOD of 0.1 pg/ml (as reported by Sanak et al.) could be enough to quantify 8-isoprostane. This is the reason why we recommend having chemical analytical instruments capable of targeting a LOD 10 times lower than the one we found, if we expect to detect these two biomarkers. This part is mentioned in the discussion (line 619), so nothing has been changed in the manuscript."
^ I understand that moving the detection limit down from 1pg/ml to 0.1 pg/ml is increasing the sensitivity by 10x, however, this is assuming the samples are in the range of 0.1 - 0.99 pg for example. Perhaps this might help the detection of both mediators in patients with asthma and/or COPD during their inflamed status, but it is unlikely to be detected in healthy volunteers.
Authors’ response: The fact that we do not detect any 8-OHdG nor 8-isoprostane in EBC samples with our analytical method (which presents a good detection limit of 0.5 or 1 pg/ml respectively) means that these levels are much lower in this matrix. Indeed, we assume in agreement with Table 3 that the lowest 8-isoprostane concentrations could be comprised between 0.2 and 0.9 pg/ml for healthy volunteers. Sanak et al. (1) reported slightly higher levels for asthmatics compared to healthy volunteers (0.32 versus 0.26 pg/ml). That is the reason why we propose to develop methods in the future able to reach detection levels around 0.1 pg/ml.
(1) Sanak, M.; Gielicz, A.; Bochenek, G.; Kaszuba, M.; Niżankowska-Mogilnicka, E.; Szczeklik, A. Targeted eicosanoid lipidomics of exhaled breath condensate provide a distinct pattern in the aspirin-intolerant asthma phenotype. Journal of Allergy and Clinical Immunology 2011, 127, 1141-1147.e1142, doi:https://doi.org/10.1016/j.jaci.2010.12.1108.
2)
"Did the authors measure levels of 8-OHdG and 8-iso in patients to check if they are confounders? (i.e.: the patients do express higher levels perhaps in the blood, but it was not detectable in EBC)
Authors’ response: We assumed that patients with asthma or COPD had a higher concentration of biomarkers in the EBC (1,2,3). Unfortunately, we did not see any difference between our volunteers (table S4). One explanation could be that our patients were not in exacerbate /inflammatory state during the collection period.
We did not collect blood because it is an invasive method and therefore undesirable for routine screening of the population."
^ Since the volunteers were obtained from the same workplace, there is a small chance the samples obtained here do not follow normal distribution or perhaps are somewhat skewed in this case.
The ability to determine if these patients express higher levels of 8-isoprostane and/or 8-OHdG systematically or in the lungs, but just not present in sufficient quantities in EBC.
If patients with asthma and/or COPD require exacerbation state in order to detect 8-isoprostane and 8-OHdG, then the detection for such a parameter would be somewhat redundant since spirometry would probably provide a better measurement in real-time, especially during exacerbation. On that note, what is the scientific rationale to measure both mediators if they are present only during exacerbation state?
Authors’ response: We think it is unlikely that the selection of our volunteers is not representative either of exposure conditions leading to increased oxidative stress or of a health condition with an inflammatory component (asthma or COPD).
Indeed, in the frame of an epidemiological study (ROBOCOP project, whose protocol is described in detail in (2)), we first analysed both 8-isoprostane and 8-OHdG oxidative stress biomarkers in EBC sample in nine workers belonging to three categories of underground subway workers: locomotive operators, security guards, and station agents. These jobs are considered the most exposed to particulate matter compared to other underground professionals. They are also supposed to represent different scenarios characterized by different levels/categories of particulate matter. The study was scheduled for six consecutive weeks, to allow a two-week follow up for every occupation, for a measure twice a day. This protocol allowed us to collect a total of 168 EBC samples. We theorized that the distribution of oxidative stress and related biomarkers in EBC would be modified in our workers depending on exposures (2, 3). However, all samples had non-detectable levels of 8-OHdG and 8-isoprostane (data not shown in the submitted manuscript), suggesting concentrations in EBC lower than 0.5 and 1pg/ml, respectively.
Based on these negative results, we conducted a second round of analysis in the same workplace by selecting seven diagnosed asthmatics and nine COPD volunteers among 303 RATP workers. We investigated if the recognized inflammatory state of these diseases would be reflected by increased 8-OHdG and/or 8-isoprostane concentrations in the EBC in these individuals. Nevertheless, similarly to the previous study, we could not detect our biomarkers of interest in our workers (supplementary material, Table S4). We therefore conclude with Reviewer 2 that the levels of these two biomarkers are below 0.5-1 pg/ml EBC.
The rationale to measure 8-isoprostane and 8-OHdG in EBC is that these two molecules are recognised biomarkers of oxidative stress. For inflammatory-based diseases such as asthma or COPD, these two molecules could have some clinical interest as early effect biomarkers. One additional advantage of measuring 8-isoprostane in EBC is that this biomarker appears related to the COPD’s severity status (GOLD stage) (4-9). Nevertheless, in our case, the calculated ratio of FEV1/FVC ratio in COPD (forced expiration to the full forced vital capacity) was adequately broad to cover different states of severity of the disease from “mild” to “moderate” (Table 2).
(2) Guseva Canu, I.; Hemmendinger, M.; Sauvain, J. J.; Suarez, G.; Hopf, N. B.; Pralong, J. A.; Ben Rayana, T.; Besançon, S.; Sakthithasan, K.; Jouannique, V.; Debatisse, A., Respiratory Disease Occupational Biomonitoring Collaborative Project (ROBoCoP): A longitudinal pilot study and implementation research in the Parisian transport company. J Occup Med Toxicol 2021, 16 (1), 22.
(3) Guseva Canu, I.; Crézé, C.; Hemmendinger, M.; Ben Rayana, T.; Besançon, S.; Jouannique, V.; Debatisse, A.; Wild, P.; Sauvain, J. J.; Suárez, G.; Hopf, N. B., Particle and metal exposure in Parisian subway: Relationship between exposure biomarkers in air, exhaled breath condensate, and urine. Int J Hyg Environ Health 2021, 237, 113837.
(4) Ashmawi, S.A.; Dewdar, I.; Mohamed, N.; Elhefny, A. Measurement of 8-isoprostane in exhaled breath condensate of patients with chronic obstructive pulmonary disease. The Egyptian Journal of Chest Diseases and Tuberculosis 2018, 67, 226, doi:10.4103/ejcdt.ejcdt_34_17.
(5) Carpagnano, G.E.; Kharitonov, S.A.; Foschino-Barbaro, M.P.; Resta, O.; Gramiccioni, E.; Barnes, P.J. Supplementary oxygen in healthy subjects and those with COPD increases oxidative stress and airway inflammation. Thorax 2004, 59, 1016-1019, doi:10.1136/thx.2003.020768.
(6) Kaźmierczak, M.; Ciebiada, M.; Pękala-Wojciechowska, A.; Pawłowski, M.; Pietras, T.; Antczak, A. Correlation of inflammatory markers with echocardiographic parameters of left and right ventricular function in patients with chronic obstructive pulmonary disease and cardiovascular diseases. Pol Arch Med Wewn 2014, 124, 290-297, doi:10.20452/pamw.2291.
(7) Ko, F.W.; Lau, C.Y.; Leung, T.F.; Wong, G.W.; Lam, C.W.; Hui, D.S. Exhaled breath condensate levels of 8-isoprostane, growth related oncogene alpha and monocyte chemoattractant protein-1 in patients with chronic obstructive pulmonary disease. Respir Med 2006, 100, 630-638, doi:10.1016/j.rmed.2005.08.009.
(8) Makris, D.; Paraskakis, E.; Korakas, P.; Karagiannakis, E.; Sourvinos, G.; Siafakas, N.M.; Tzanakis, N. Exhaled Breath Condensate 8-Isoprostane, Clinical Parameters, Radiological Indices and Airway Inflammation in COPD. Respiration 2008, 75, 138-144, doi:10.1159/000106377.
(9) Antus, B.; Harnasi, G.; Drozdovszky, O.; Barta, I. Monitoring oxidative stress during chronic obstructive pulmonary disease exacerbations using malondialdehyde. Respirology 2014, 19, 74-79, doi:10.1111/resp.12155.